# OpenReview forum: "Traceback of Poisoned Texts in Poisoning Attacks to Retrieval-Augmented Generation"
_ACM.org/TheWebConf/2025/Conference — WWW 2025 Poster_

### Official Review · Reviewer_HmNp · 2024-11-25

**Novelty:** 6
**Technical Quality:** 6

**Review:**

The RAGForensics system proposed in this paper is an important improvement to the security of existing RAG systems. It introduces a traceability mechanism within the RAG framework, effectively addressing poisoning attacks on external knowledge databases. The innovation of the paper lies in the fact that RAGForensics can trace and identify potentially poisoned texts without relying on model gradients, overcoming the limitations of previous inference-time defense mechanisms. Empirical evaluation results show that RAGForensics performs excellently across multiple datasets, effectively countering advanced poisoning attacks. This work opens up a new direction for research on securing RAG systems against poisoning, offering significant practical value and potential.

**Questions:**

1.There are some grammatical errors in the text, and the clarity of expression needs to be improved overall.

**Ethics Review Flag:**

Yes

**Reviewer Confidence:**

3: The reviewer is confident but not certain that the evaluation is correct

**Scope:**

4: The work is relevant to the Web and to the track, and is of broad interest to the community

---

### Official Review · Reviewer_LJkp · 2024-11-30

**Novelty:** 5
**Technical Quality:** 5

**Review:**

Summary:
This work proposes a new approach in addressing poisoning attack in a relatively new area --- poisoning the knowledge base of large language models, by tracing the poisoned text with higher accuracy. The approach is evaluated on 3 query datasets labelled with correct answers against 3 poisoning attacks and 2 adaptive attacks.

Pros:
+ The application area of poisoning attack in relatively unexplored.
+ The approach shows promising performance.
+ The evaluation considers adaptive attacks and defence strategies.
Cons:
-	The experiment is small with only 50 queries selected as the final test data, which hinders the robustness of the performance.
-	The proposed approach relies heavily on assumptions of attacks, and relies on LLM to defeat attack for LLM, lacking insights on how and why they work.
-	It seems more lab experiments, lacking discussion on practicability like testing in the wild.

Significance: The targeted problem --- poisoning of knowledge base is an emerging problem and getting increasingly important with the application of LLM. The work demonstrates that the proposed approach can effectively identify poisoned texts in knowledge base.

Novelty: The targeted research area is relatively new. The angle to address the problem is new. The technique is weak in innovation.

Technical Quality: The approach has two main parts: narrowing scop and identifying poisoned text. The narrowing scope part is based on the assumption that attacker crafts poisoned text to maximize the similarity with the targeted query, which has limitations. In addition to the limitation stated in the paper that it is vulnerable for non-targeted attacks, it may also suffer limitations on targeted attacks that do not use the same assumption. The high performance of the experiments may be because of the 3 targeted attacks use the same strategy. It would be clearer if there is a test on the pervasive of this assumption in the poisoning attack strategies beyond RAG.
The identifying poisoned text is based on LLM with explicit instruction and chain-of-thought. It is worth discussing the pervasive of the application of these instructions and the prompts, as well as providing in-depth insights to understand why they work. Using LLM to defend poisoning attacks on LLM, means that poisoning attack may be applied for the LLM defence to deceive the defence. The discussed adaptive attack does not cover this case.
In the experiments, the poisoned attack is lab generated. This is reasonable as the performance metrics require ground truth. But it would be more impactful if there are experiments in the wild to actually detect poisoning attack in the deployed LLMs.

Clarity: The work is clearly presented in general. A downside is that the discussion on non-poisoned feedback and the Benight Texts Enhancement is in appendix, while they appear in two (out of the three) stated contributions.

Detailed comments:
Page 1: What is “inference -time defenses”?
Page 3 Section 3.1 “the attacker has full knowledge of the texts in the database and direct access to the parameters of both the retriever and the LLM”. Is this assumption necessary, practical or pervasive in the attacks?
Page 4. What is R?
Page 6. Section 5.1.6 A justification of the setting is needed, i.e., why choosing the RAG, Attack and traceback system settings as stated.
Page 8: For the non-poisoned attack, are they common in the experiment? An analysis or example would be helpful in understanding how important the problem is.

Typos:
Page 2: we discuss -> We discuss

**Questions:**

- Can large scale of experiment feasible?
- Is this approach practical if tested in the wild? What would be the challenges to deploy the approach?
- Are there ways to lift the assumptions to enhance the wide adoption of the approach?

**Reviewer Confidence:**

3: The reviewer is confident but not certain that the evaluation is correct

**Scope:**

3: The work is somewhat relevant to the Web and to the track, and is of narrow interest to a sub-community

---

### Official Review · Reviewer_bqcR · 2024-11-30

**Novelty:** 5
**Technical Quality:** 5

**Review:**

Aiming at the problem that the RAG system is vulnerable to poisoning attacks, a traceability system named RAGForensics is proposed. The system can accurately identify the poisoned text from the knowledge database through the observed attacks. Experiments show that the system is effective, universal and robust. However, the system still has some limitations.

**Questions:**

1、In Introduction, please explain why tracing toxic texts is more effective than establishing complex defense mechanisms. “In light of this challenge, we argue that it is more practical for RAG system providers to focus on tracing poisoned texts rather than building sophisticated defenses to prevent adversarial manipulation of the LLM….” How can this comparison be proved?
2、Please provide necessary quotations and reasonable explanations to explain the reasons for "the attacker's strategy", which are in sections 4.2 and 4.3 respectively.
3、Lack of explanation of experimental parameter settings. For example, in RAG settings, why do you take the Top-5 most relevant texts in the knowledge database for each query? Is it verified by hyperparameter experiment?
4、The 6 baseline models extend the backtracking method of neural network poisoning attack and adapt to the common attack measures in RAG system. Is there any basis or quotations for this practice and whether the baseline experimental results obtained are reliable?

**Reviewer Confidence:**

3: The reviewer is confident but not certain that the evaluation is correct

**Scope:**

3: The work is somewhat relevant to the Web and to the track, and is of narrow interest to a sub-community

---

### Official Review · Reviewer_SpJG · 2024-12-01

**Novelty:** 4
**Technical Quality:** 4

**Review:**

This paper presents the first traceback system in RAG, RAGForensics, which traces poisoned texts from the knowledge database. RAGForensics includes two phases, i.e., narrowing the space of potentially poisoned texts and accurately identifying them without requiring access to model gradients. The evaluation on multiple datasets shows RAGForensics’s effectiveness against state-of-the-art and adaptive poisoning attacks.

**Strengths**:
+ Addresses an important and timely problem.
+ Provides a clear and well-structured description of the problem scenario and approach, making the paper easy to follow.

**Weaknesses**:
+ Nolvety: The fundamental weakness of the paper is that the approach proposed is too simple.
+ Analysis:Lack of in-depth analysis of experimental results

Detailed comments :

To be specific, the authors claim that RAGForensicssystem operates in two key phases: narrowing the scope of potential poisoned texts (Section 4.2) and accurately identifying them (Section 4.3). In the narrowing phase, the system retrieves a subset of texts from the database that are most likely to be poisoned, reducing the identification scope from millions to just a few dozen texts.

In Section 3.1, the authors says that they consider a practical scenario where the RAG owner uses a closed-source retriever and LLM that outperform open-source alternatives. As a result, the traceback system cannot access their internal parameters but can query them directly. Hence, they assume that the traceback system has collected each targeted query and its corresponding RAG output involved in the poisoning attack.

However, in Section 4.2, I found that the authors leverage the attacker’s strategy and then only employ the RAG retriever to select the top-𝐾 relevant texts for the query. Hence, I have a question that what does RAGForensicssystem contribute in the first phase? It seems that RAGForensicssystem only need to directly return the RAG result to the LLM. I did not clearly see the main contribution or design in this phase.  It seems that the description in Introduction is an overclaim of RAGForensicssystem.

For Phase 2, in Section 4.3, the identification phase leverages an LLM to precisely detect the poisoned texts within this subset, eliminating the need for gradient calculations.
However, there is no clear motivation of leveraging LLM to detect the poisoned texts. Why LLM is the most suitable for your problem, and can other techniques detect them precisely? Even though this paper is the first work to explore a traceback of poisoning texts in RAG, the approach heavily relies on the LLMs. The usage of prompts is a bit simple. Authors claim the approach incorporate the chain-of-thought (CoT) approach by including the instruction, “Let’s think step by step”. How effective is this instruction for contemporary LLMs? Can you compare with more advanced CoT approaches?
To sum up, I think that the paper should explicitly discuss its limitations and position its contributions more modestly.


In the experiments, the authors initially select 100 queries from PoisonedRAG for each dataset, but only 50 targeted queries are chosen as the final test set. The rationale behind this selection process is unclear and requires clarification.

There is a noticeable lack of in-depth analysis of the experimental results. For instance, when evaluating the impact of the number of poisoned texts injected into the knowledge database, the paper should provide more detailed insights into the observed trends.
Why does RAGForensics consistently outperform other methods under the InstruInject attack on the MS-MARCO dataset but not on NQ or HotpotQA? Further discussion on this discrepancy is needed.


The term "Benign Texts Enhancement (BTE)" is introduced as an abbreviation without first providing its full name. Ensure all abbreviations are defined upon first use for better clarity.

**Questions:**

Please check my concerns in the comments.

**Reviewer Confidence:**

3: The reviewer is confident but not certain that the evaluation is correct

**Scope:**

3: The work is somewhat relevant to the Web and to the track, and is of narrow interest to a sub-community